# Assessment of Quadriceps Muscle in Advanced Knee Osteoarthritis and Correlation with Lower Limb Alignment

**DOI:** 10.3390/medicina60121983

**Published:** 2024-12-02

**Authors:** Ki-Cheor Bae, Eun-Seok Son, Chang-Jin Yon, Jubin Park, Du-Han Kim

**Affiliations:** Department of Orthopedic Surgery, Keimyung University Dongsan Hospital, Keimyung University School of Medicine, Daegu 42601, Republic of Korea; bkc@dsmc.or.kr (K.-C.B.); esson@dsmc.or.kr (E.-S.S.); poweryon@dsmc.or.kr (C.-J.Y.); 230514@dsmc.or.kr (J.P.)

**Keywords:** knee, osteoarthritis, quadriceps, varus deformity, muscle atrophy

## Abstract

*Background and Objectives*: Despite extensive studies of the role of quadriceps and quadriceps/hamstring balance in knee osteoarthritis (OA), the roles of the vastus intermedius, medialis, and lateralis in OA remain unclear. The purpose of this study was to investigate the relationship of lower limb alignment and the ratio of the quadriceps femoris muscle to the knee extensor muscle. *Materials and Methods*: This study included 50 patients with advanced knee OA (Kellgren/Lawrence grade of 3 or 4) and 25 healthy control persons between June 2021 and May 2022. The osteoarthritis grade and anatomical tibiofemoral angle were measured based on plain radiography and scanography. All participants were divided into normal (0~5°), mild varus (5°~10°), and severe varus (>10°) groups. Using MRI, muscle size was determined by calculating the cross-sectional area (CSA) of the total quadriceps (rectus femoris, vastus intermedius, vastus medialis, and vastus lateralis) and its components. *Results*: The CSA ratio of the vastus lateralis was significantly smaller in the severe varus group than in the normal or mild varus groups. There was a significant positive correlation between the mechanical tibiofemoral angle and vastus lateralis CSA (*ρ* = 0.282, *p* = 0.014) and between the anatomical tibiofemoral angle and vastus lateralis CSA (*ρ* = 0.294, *p* = 0.011). There was a significant negative correlation between the mechanical tibiofemoral angle and vastus intermedius CSA (*ρ* = −0.263, *p* = 0.023) and between the anatomical tibiofemoral angle and vastus intermedius CSA (*ρ* = −0.243, *p* = 0.036). *Conclusions*: Patients with severe varus alignment exhibited vastus lateralis atrophy. This study highlights vastus lateralis atrophy in severe varus alignment, though causality between atrophy and varus knee OA remains uncertain. We think that patients with severe varus may require strengthening exercises focused on the vastus lateralis before and after surgery for alignment correction.

## 1. Introduction

Quadriceps muscle dysfunction is a common feature in knee osteoarthritis (OA) and is associated with functional decline and disease progression. However, patient satisfaction rates lower than 80% after conventional total knee arthroplasty have been reported in the literature [1,2,3,4]. A typical feature of knee osteoarthritis (OA) is knee extensor weakness, which is associated with the development of symptomatic knee OA as well as functional decline over time in people with knee OA [5]. Knee extensor muscle condition has been studied in patients with knee osteoarthritis to determine its association with varying degrees of functional impairment and disease progression. Several studies have suggested that the quadriceps muscle seems to be related to lower risk of symptomatic OA, but not radiographic OA [5]. While quadriceps muscles have been studied, their specific contributions to OA pathogenesis, particularly in relation to lower limb alignment, remain less established. 

In particular, the vastus medialis (VM) is being studied in various ways in knee-related conditions such as patellofemoral pain syndrome and knee osteoarthritis [6,7]. Patients exhibited atrophy of the VM in patellofemoral pain syndrome. Pattyn et al. found that the cross-sectional area (CSA) of VM was significantly smaller in the patients’ group. A tendency was also noted for the patellofemoral pain syndrome group toward a smaller CSA of the total quadriceps at the midthigh level [6]. While VM has been associated with patellofemoral pain syndrome, its role in knee OA, particularly in conjunction with other quadriceps components, warrants further investigation. According to the knee OA study, increased VM size was associated with reduced knee pain and structural improvements in knee OA [3,4,7,8]. Conversely, Pan et al. suggested that the ratio between vastus lateralis (VL) and VM cross-sectional area may influence the progression of OA [9].

However, there are few studies of the relationship between lower limb alignment and knee extensor muscle volume in knee OA. The purpose of this study was to investigate the relationship between lower limb alignment and the ratio of the quadriceps femoris muscle to the knee extensor muscle using magnetic resonance imaging (MRI). We hypothesize that lower limb alignment would be correlated with the ratio of the quadriceps femoris muscle.

## 2. Materials and Methods

### 2.1. Participants

We performed a retrospective review of medical records for patients with knee OA from June 2021 and May 2022. The diagnosis of knee OA was established using patient medical history, plain radiographs, and MRI. This study included a total of 75 patients, comprising 50 patients with knee OA and 25 healthy individuals who had never had knee pain before.

Patients who met the following criteria were included in the study: (1) >20 years old, and (2) patients with confirmed symptomatic knee OA with a Kellgren/Lawrence (KL) grade of 3 or 4 [10]. Exclusion criteria included patients who (1) had no preoperative MRI or a poor-quality image that was difficult to measure, (2) had concomitant fractures, or (3) had a previous history of knee surgery or trauma.

### 2.2. Radiographic Evaluations

Radiographic images were acquired using an SH3 system (DK Medical, Seoul, Republic of Korea). Knee OA was confirmed through anterior–posterior and lateral weight-bearing radiographs, as well as non-weight-bearing skyline radiographs. An experienced orthopedic surgeon utilized these images to grade bilateral tibiofemoral OA based on the KL criteria. Participants were included in the knee OA group if they were classified as grade III or IV (unilateral or bilateral) on the KL scale and exhibited clinical symptoms. Participants were included in the control group if their knees were classified as grade 0 or I on the KL scale and were asymptomatic.

All participants underwent conventional scanography. Patients were positioned to stand with equal weight distribution on both lower extremities without an assistive device, with both patellae pointing anteriorly and feet straight, parallel to each other. The patient-to-tube distance was typically 101 cm. Three separate 35 × 9 × 43 cm cassettes were used to obtain three separate images centered over the hip, knee, and ankle joints. Based on scanography alignment, the hip–knee–ankle (HKA) angle was measured and defined as the angle between the mechanical axes of the femur and tibia. All participants were divided into normal (Group I), mild varus (Group II), and severe varus (Group III) groups. Group I was defined as between 5 degrees of valgus and 5 degrees of varus, Group II was defined as between 5 and 10 degrees of varus, and Group III was defined as >10 degrees of varus [11].

### 2.3. Thigh Muscle Cross-Sectional Area Measurement

A consecutive series of MRI scans of each patient’s thighs was acquired using a 1.5T MRI system (Siemens Magnetom Avanto System; Siemens Medical, Erlangen, Germany). Close collaboration with the radiology department was maintained throughout the study to ensure precise measurements. Oblique sagittal T2 TSE MRI scans, perpendicular to the long axis of the scapular body, were obtained with a slice thickness of 3 mm. The contours of the rectus femoris (RF), vastus intermedius (VI), vastus medialis (VM), and vastus lateralis (VL) muscles were manually traced using PACS software (Picture Archiving and Communication System, DICOM version 3.0 INFINITT, Infinitt Healthcare, Seoul, Republic of Korea). As illustrated in Figure 1, the borders of the four muscle groups were delineated in the mid-thigh view on the axial plane. The software then automatically calculated the cross-sectional area (CSA) for each muscle. Subsequently, the percentage of each muscle relative to the total quadriceps muscles (rCSA) was determined. Two knee-specializing orthopedic surgeons, who worked in the territory university hospital (*** and ***), independently performed the image measurements. Interobserver reliability was reassessed through a randomized analysis conducted three months later.

### 2.4. Statistical Analysis

Statistical analysis was performed using SPSS 24.0 software. Two methods were used to assess the measurement reliability. The Mann–Whitney test was used to evaluate the difference in rCSA between the OA patients and the control group. The mean difference of the rCSA between each alignment group was estimated using the Kruskal–Wallis test. Furthermore, the Spearman correlation coefficient was used to examine the correlation between the lower limb alignment and CSA. The test–retest reliability was evaluated initially; subsequently, internal consistency reliability was assessed using the Bland and Altman method for calculating inter- and intraclass correlation (ICC). A threshold for statistical significance was established at *p* < 0.05. Results with *p*-values below this threshold were deemed statistically significant, whereas those with *p*-values above this threshold were not.

## 3. Results

Participant characteristics are shown in Table 1. Age and BMI (mean ± SD) were 69.3 ± 6.7 years and 26.4 ± 3.2 kg/m^2^ in the OA group, and 44.8 ± 15.1 years and 27.6 ± 4.7 in control group, respectively.

### 3.1. OA Grade Differences in Thigh Muscle CSA Measurements

The CSA of the total quadriceps and each component was significantly smaller in the OA group than in the control group. Additionally, we analyzed the ratio of percentages between each component. As a result, no significant differences of the CSA ratio between the knee OA patients and the control group were found (Table 2).

### 3.2. Muscle CSA in Relation to Knee Alignment

Table 3 shows participant characteristic data classified according to mechanical alignment. The CSA ratio of the vastus lateralis was significantly smaller in the severe varus group than in the normal (Z = −2.452, *p* = 0.014) or mild varus groups (Z = −2.251, *p* = 0.011) (Table 4). There was a significant positive correlation between the mechanical tibiofemoral angle and vastus lateralis CSA (*ρ* = 0.282, *p* = 0.014), and between the anatomical tibiofemoral angle and vastus lateralis CSA (*ρ* = 0.294, *p* = 0.011). There was a significant negative correlation between the mechanical tibiofemoral angle and vastus intermedius CSA (*ρ* = −0.263, *p* = 0.023), and between the anatomical tibiofemoral angle and vastus intermedius CSA (*ρ* = −0.243, *p* = 0.036).

## 4. Discussion

Our study examined differences in thigh muscle CSA among participants with knee OA and a control group. While overall quadriceps CSA was reduced in OA patients, the disproportionate atrophy of the vastus lateralis in the severe varus group suggests a targeted impact of knee alignment on muscle morphology.

Notably, the CSA ratio of the vastus lateralis was considerably smaller in the severe varus group compared to the normal or mild varus groups. Additionally, there were significant correlations found between knee alignment and CSA. There was a positive correlation with the vastus lateralis and a negative correlation with the vastus intermedius, highlighting how changes in knee mechanics are associated with muscle morphology in knee OA.

The quadriceps femoris muscle is commonly associated with tibiofemoral OA and is a critical factor in determining disability [12]. Thomas et al. found that women with radiographic OA had 22% less quadriceps strength compared to women without OA, aligning with various reports of isometric and isokinetic knee extension torque deficits ranging from 20% to 40% [13,14,15]. Adequate quadriceps strength in knee OA patients is essential for performing daily activities, and strengthening this muscle has been shown to improve physical function in those affected by the disease [12,16,17]. Pietrosimone et al. reported that individuals with high physical activity levels had significantly greater quadriceps strength compared to those with lower activity levels, indicating a positive association between quadriceps strength and physical activity [18]. Greater quadriceps muscle strength is also linked to a lower risk of developing symptomatic knee OA [19]. However, evidence supporting the quadriceps muscle’s significant role in the incidence of radiographic knee OA remains limited.

There are several methods used to evaluate the quadriceps muscle change in patients with osteoarthritis [20,21,22]. Using ultrasonography to measure muscle thickness and echo intensity in various lower extremity muscles, Taniguchi et al. measured muscle quantity and quality in 21 women with knee OA and 23 healthy individuals. Their results indicated that muscle changes associated with OA, such as decreased muscle thickness and increased echo intensity, are more pronounced as OA severity increases and vary among different muscles, particularly in the vastus medialis and intermedius [21]. Segal et al. conducted their study using dual-energy X-ray absorptiometry. They found that, contrary to previous beliefs about the protective role of quadriceps strength, greater thigh muscle mass does not protect against the development or worsening of knee OA or joint space narrowing. Instead, higher specific strength in the knee extensors significantly lowers the risk for symptomatic knee OA and joint space narrowing. This result suggests that neuromuscular activation and muscle physiology may be more crucial factors to consider than muscle mass alone [23].

Research suggests that muscle atrophy and fatty infiltration are common in the quadriceps of knee OA patients, often with disproportionate atrophy in the vastus medialis [7,8,11,24]. Van der Noort et al. found that a higher percentage of non-contractile tissue in the VM muscle is linked to weaker muscle strength and longer performance times in mobility tests in patients with knee OA measured by MRI. Their results also showed that women with a high BMI were more at risk for having a high amount of non-contractile tissue [24]. Fink et al. performed histopathological examination of the vastus medialis muscle. In their study, patients with end-stage OA revealed widespread atrophy and changes indicative of neurogenic muscular atrophy and pain-associated disuse, highlighting the multifactorial nature of muscle deterioration in osteoarthritis. Their findings suggest a complex interaction between structural muscle changes and clinical symptoms like leg axis deviation [11]. Our findings of targeted vastus lateralis atrophy in severe varus alignment complement previous studies showing muscle-specific changes in OA, suggesting that muscle morphology may be influenced by joint mechanics. We are also not sure which came first. However, we think that this group of patients may require strengthening exercises focused on the vastus lateralis before and after surgery for alignment correction.

Our study had several limitations. First, the retrospective nature of this study limited the ability to establish causality between muscle cross-sectional area changes and knee osteoarthritis progression. The second limitation was measurement bias. While muscle CSA measurements were performed by experienced orthopedic surgeons, manual tracing and reliance on clinical MRI may have introduced variability that could affect the precision and repeatability of the measurements. If automated three-dimensional reconstruction had been used as a measurement method, accuracy may have been even higher. Third, this study sample was comprised only of patients with advanced knee osteoarthritis and a control group, which may not be representative of all stages of the disease or broader demographic characteristics. We also did not perform an age- and sex-matched study because we wanted to compare an old-age OA group with a young healthy group.

## 5. Conclusions

When compared to healthy individuals, patients with advanced knee OA had extensor muscle atrophy. However, there was no difference in the ratio of each component. Patients with severe varus alignment exhibited vastus lateralis atrophy. Although it is not clear whether this unbalanced atrophy is a result or a cause of varus knee OA, our results suggest that vastus lateralis atrophy may contribute to the severity of varus knee OA. To establish the cause–effect relation of vastus lateralis atrophy and varus knee OA, longitudinal and prospective studies on the causal relationship between muscle atrophy and knee alignment are needed.

## Figures and Tables

**Figure 1 medicina-60-01983-f001:**
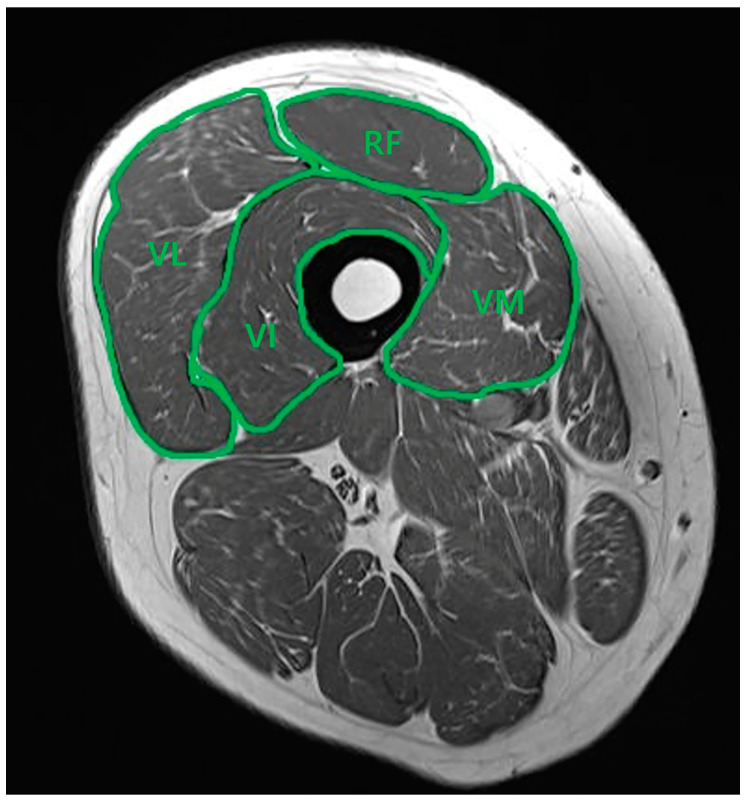
Axial T2 TSE MRI. Measurement of the cross-sectional area of the quadriceps muscle in mid-thigh view. Rectus femoris (RF), vastus intermedius (VI), vastus medialis (VM), and vastus lateralis (VL).

**Table 1 medicina-60-01983-t001:** Demographic data of participants classified according to OA status.

	OA Group (*n* = 50)	Control Group (*n* = 25)	*p*-Value
Age (years)			0.000 *
Mean ± SD	69.3 ± 6.7	44.8 ± 15.1	
Gender			0.008 *
Male, *n* (%)	9 (18.0%)	13 (52.0%)	
Female, *n* (%)	41 (82.0%)	12 (48.0%)	
Height (cm)			0.000 *
Mean ± SD	154.9 ± 9.6	165.9 ± 8.4	
Weight (kg)			0.000 *
Mean ± SD	63.3 ± 9.0	76.3 ± 16.1	
BMI (kg/m^2^)			0.247
Mean ± SD	26.4 ± 3.2	27.6 ± 4.7	
Side			0.624
Right, *n* (%)	23 (46.0%)	13 (52.0%)	
Left, *n* (%)	27 (54.0%)	12 (48.0%)	
Alignment classification			0.000 *
Normal, *n* (%)	13 (26.0%)	23 (92.0%)	
Mild varus, *n* (%)	15 (30.0%)	2 (8.0%)	
Severe varus, *n* (%)	22 (44.0%)	0 (0%)	
OA grade (KL)			0.000 *
KL 0, *n* (%)	0 (0%)	4 (16.0%)	
KL 1, *n* (%)	0 (0%)	11 (44.0%)	
KL 2, *n* (%)	0 (0%)	10 (40.0%)	
KL 3, *n* (%)	5 (10.0%)	0 (0%)	
KL 4, *n* (%)	45 (90.0%)	0 (0%)	

OA: osteoarthritis, SD: standard deviation, BMI: body mass index, KL: Kellgren/Lawrence. * statistically significant.

**Table 2 medicina-60-01983-t002:** CSA ratio of the quadriceps components.

	OA Group (*n* = 50)	Control Group (*n* = 25)	*p*-Value
RF	10.52 ± 2.15	10.94 ± 1.92	0.408
VI	33.14 ± 3.48	32.26 ± 4.27	0.338
VM	22.39 ± 2.77	21.67 ± 3.39	0.327
VL	33.94 ± 2.65	35.13 ± 4.17	0.203

CSA: cross-sectional area, RF: rectus femoris, VI: vastus intermedius, VM: vastus medialis, VL: vastus lateralis.

**Table 3 medicina-60-01983-t003:** Demographic data of participants classified according to mechanical alignment.

	Group I (*n* = 36) Normal Alignment	Group II (*n* = 17) Mild Varus Alignment	Group III (*n* = 22) Severe Varus Alignment	*p*-Value	
Age (years)				0.000 *	I < II, III
Mean ± SD	51.2 ± 17.4	66.4 ± 10.5	69.3 ± 7.7		
Gender				0.247	
Male, *n* (%)	14 (38.9%)	3 (17.6%)	5 (22.7%)		
Female, *n* (%)	22 (61.1%)	14 (82.4%)	17 (77.3%)		
Height (cm)				0.008 *	I < III
Mean ± SD	163.5 ± 9.8	155.4 ± 8.9	154.6 ± 10.6		
Weight (kg)				0.028 *	I < III
Mean ± SD	73.2 ± 14.6	65.5 ± 12.8	62.1 ± 10.6		
BMI (kg/m^2^)				0.564	
Mean ± SD	27.3 ± 4.1	27.1 ± 4.0	26.0 ± 3.7		
Side				0.594	
Right, *n* (%)	20 (55.6%)	5 (29.4%)	11 (50.0%)		
Left, *n* (%)	16 (44.4%)	12 (70.6%)	11 (50.0%)		
OA grade (KL)				0.000 *	
KL 0, *n* (%)	4 (11.1%)	0 (0%)	0 (0%)		
KL 1, *n* (%)	10 (27.8%)	1 (5.9%)	0 (0%)		
KL 2, *n* (%)	9 (25.0%)	1 (5.9%)	0 (0%)		
KL 3, *n* (%)	2 (5.6%)	1 (5.9%)	2 (9.1%)		
KL 4, *n* (%)	11 (30.6%)	14 (82.4%)	20 (90.9%)		

OA: osteoarthritis, SD: standard deviation, BMI: body mass index, KL: Kellgren/Lawrence. * statistically significant.

**Table 4 medicina-60-01983-t004:** CSA ratio of the quadriceps components in each alignment group.

	Group I (*n* = 36) Normal Alignment	Group II (*n* = 17) Mild Varus Alignment	Group III (*n* = 22) Severe Varus Alignment	*p*-Value	
RF	10.86 ± 2.12	9.92 ± 2.17	10.90 ± 1.86	0.251	
VI	31.88 ± 3.71	33.05 ± 2.62	34.27 ± 4.21	0.058	
VM	22.31 ± 3.22	21.96 ± 3.00	22.04 ± 2.70	0.907	
VL	34.94 ± 3.67	35.06 ± 3.00	32.78 ± 2.14	0.027 *	I, II > III

RF: rectus femoris, VI: vastus intermedius, VM: vastus medialis, VL: vastus lateralis. * statistically significant.

## Data Availability

The data presented in this study are available on request from the corresponding author due to privacy concerns.

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
