# Peer review of "Assessment of Quadriceps Muscle in Advanced Knee Osteoarthritis and Correlation with Lower Limb Alignment"

_medicina, 2024, doi:10.3390/medicina60121983_

Round 1

Reviewer 1 Report

Comments and Suggestions for Authors

This manuscript investigates the relationship between lower limb alignment and the ratio of quadriceps femoris muscle in knee extensor muscles. The study finds that the cross-sectional area (CSA) of the vastus lateralis positively correlates with mechanical and anatomical tibiofemoral angles. Conversely, a negative correlation is observed between the CSA of the vastus intermedius and the mechanical and anatomical tibiofemoral angles. This is an interesting observation.

Specific comments:

  1. It is imperative that a native English speaker is recruited to refine the language of the manuscript, ensuring that the findings are effectively communicated and the study's significance is fully appreciated.
  2. An ethical statement should be provided given the nature of the human subjects involved.
  3. Information about who diagnoses the severity of the OA patients should be provided.
  4. Line 96: Information on the orthopedic surgeons is missing.
  5. Significant differences are noted between OA and Control groups regarding age, gender, height, and weight. How these factors influence the conclusion should be discussed as well.
  6. Beyond noting the correlation, the meaning of these correlations is not fully discussed, which significantly reduces the importance of this study.
Comments on the Quality of English Language
  1. It is imperative that a native English speaker is recruited to refine the language of the manuscript, ensuring that the findings are effectively communicated and the study's significance is fully appreciated.

Author Response

Comments and Suggestions for Authors

This manuscript investigates the relationship between lower limb alignment and the ratio of quadriceps femoris muscle in knee extensor muscles. The study finds that the cross-sectional area (CSA) of the vastus lateralis positively correlates with mechanical and anatomical tibiofemoral angles. Conversely, a negative correlation is observed between the CSA of the vastus intermedius and the mechanical and anatomical tibiofemoral angles. This is an interesting observation.

Specific comments:

  1. It is imperative that a native English speaker is recruited to refine the language of the manuscript, ensuring that the findings are effectively communicated and the study's significance is fully appreciated.

è Thank you for your comment. We asked to English editing company and revised our manuscript. We added the approved certificate.

An ethical statement should be provided given the nature of the human subjects involved.

è Thank you for your comment. We added IRB document.

  1. Information about who diagnoses the severity of the OA patients should be provided.

è Thank you for your comment. We defined the severity of OA using KL grading system (X ray). KL grade 3 and 4 was defined as the knee OA group. Please see “2.1. Subjects” section.

  1. Line 96: Information on the orthopedic surgeons is missing.

è Thank you for your comment. We added more information about orthopedic surgeons.

  1. Significant differences are noted between OA and Control groups regarding age, gender, height, and weight. How these factors influence the conclusion should be discussed as well.

è Thank you for your comment. We selected control group participants who have no history of knee surgery and treatment. We want to know the results of the comparison between the old OA group and young healthy group. For this reason, we did not sex and age matching process. We added this content in the “limitation” section.

  1. Beyond noting the correlation, the meaning of these correlations is not fully discussed, which significantly reduces the importance of this study.

è Thank you for your comment. Our hypothesis is that lower limb alignment would be correlated with the quadriceps muscle ratio. There are several studies which analyze the relationship between knee OA and muscle. We discussed these articles, however, there is no paper that focused on the alignment. As your comment, we added more contents to emphasize our results in line 187-189.

Comments on the Quality of English Language

  1. It is imperative that a native English speaker is recruited to refine the language of the manuscript, ensuring that the findings are effectively communicated and the study's significance is fully appreciated.

Reviewer 2 Report

Comments and Suggestions for Authors

Peer Review Report for "Quadriceps Muscle Assessment in Patients with Advanced Knee Osteoarthritis versus Normal Knee Anatomy: Correlation with Lower Limb Alignment"   

Title: While the title effectively conveys the study's focus, "Quadriceps Muscle Assessment" could be more specific. Consider specifying which components of the quadriceps (e.g., vastus lateralis, vastus intermedius) are being assessed. A more precise title could be, "Assessment of Vastus Lateralis and Intermedius in Advanced Knee Osteoarthritis and Correlation with Lower Limb Alignment."   

Abstract: The abstract is generally clear but could benefit from a more succinct presentation of the results and implications. The sentence, "Although it is not clear whether this unbalanced atrophy is a result or a cause of varus knee OA..." could be revised for better clarity. Consider rephrasing to: "The study highlights vastus lateralis atrophy in severe varus alignment, though causality between atrophy and varus knee OA remains uncertain."   

Also add practical implication of this study in the conclusion of the abstract.

Introduction

First paragraph (lines 31-40): The opening sentence, "Total knee arthroplasty is a well-established surgery for patients with advanced knee osteoarthritis (OA)," is somewhat disconnected from the main focus of your study. Consider replacing it with a sentence that directly introduces the quadriceps' role in knee OA, such as: "Quadriceps muscle dysfunction is a common feature in knee osteoarthritis (OA) and is associated with functional decline and disease progression."   

Lines 38-39: You mention that "the roles of the vastus intermedius, medialis, and lateralis in OA remain unclear despite extensive studies." However, later in the introduction, you cite studies that suggest a role for these muscles in OA. This creates a minor contradiction. You might want to clarify that while these muscles have been studied, their specific contributions to OA pathogenesis, particularly in relation to lower limb alignment, remain less established.   

Lines 41-50: The discussion of vastus medialis (VM) atrophy in patellofemoral pain syndrome (PFPS) is informative, but the connection to knee OA needs to be made clearer here. You might want to introduce a linking sentence to bridge the gap between PFPS and knee OA, such as: "While VM atrophy has been associated with patellofemoral pain, its role in knee OA, particularly in conjunction with other quadriceps components, warrants further investigation."  

 Lines 46-48: The repetition of the phrase "According to the knee OA study, increased VM size was associated with reduced knee pain and beneficial structural changes at the knee," should be avoided. Consider merging the two instances into one concise sentence: "Increased VM size has been associated with reduced knee pain and structural improvements in knee OA."  

Lines 49-50: The sentence, "Conversely, another author suggests that the vastus lateralis (VL) and VM balance might play an important role in OA pathogenesis," is vague. Specify which study you’re referring to and provide a clearer explanation of how this balance could influence OA. A more precise sentence could be: "Pan et al. (2011) suggested that the ratio between vastus lateralis and vastus medialis cross-sectional area may influence the progression of OA."   

Methods   

Line 56: replace "Subjects" by "Participants"   

Lines 55-65: The inclusion and exclusion criteria are clearly presented. However, the criteria for defining "healthy individuals" (control group) is not entirely clear. Were the control participants free of any knee symptoms, or were they simply classified as KL grade 0 or 1? Clarifying this distinction would strengthen your methodology transparency.  

 Lines 74-82 (Radiographic Evaluations): The description of how participants were divided into alignment groups is well-detailed. However, it would be beneficial to explain why the specific cutoffs of 5° and 10° were chosen for normal, mild, and severe varus. Were these thresholds based on prior literature or clinical guidelines? A brief justification here would improve the rationale.   Results Table 1 (Demographic Data): The comparison between the OA and control groups is informative, but the significant age difference (approximately 25 years) between the groups is a potential confounder. A possible question for the authors: Why was age not controlled for in the analysis, given its potential influence on muscle atrophy and knee alignment? You may want to mention the need for age-matched controls in future studies to mitigate this issue.   

Lines 121-127 (CSA Ratio): The statement, "no significant difference of the CSA ratio between the knee OA patients and the control group were found," is somewhat surprising given the study's focus. Could the authors provide an explanation for this finding? Specifically, is it possible that the significant reduction in total CSA in the OA group masks potential differences in the relative proportions of the quadriceps components? This discrepancy should be acknowledged and discussed.  

 Lines 130-135 (Muscle CSA and Alignment): The finding that vastus lateralis CSA was significantly smaller in the severe varus group is well-highlighted. However, the clinical significance of this result is not fully explored. A question for the authors: How might vastus lateralis atrophy contribute to the progression of varus deformity in knee OA? Could strengthening the vastus lateralis potentially mitigate further varus alignment? A brief discussion on the potential therapeutic implications would be valuable.   

Discussion 

Lines 143-175 (General Discussion of Results): Your discussion of quadriceps atrophy in knee OA patients is thorough but could benefit from a more focused interpretation of the key findings. For example, the sentence, "Although the ratios of muscle CSA were similar across both groups, OA group participants showed significantly reduced CSA of the total quadriceps," could be more directly tied to the clinical relevance of vastus lateralis atrophy. Consider rephrasing to: "While overall quadriceps CSA was reduced in OA patients, the disproportionate atrophy of the vastus lateralis in the severe varus group suggests a targeted impact of knee alignment on muscle morphology."   

Lines 176-185: The discussion on muscle atrophy and fatty infiltration is relevant but could be better aligned with your own findings. Specifically, you might want to emphasize how your results contribute to the understanding of muscle-specific atrophy in knee OA. For instance, you could add: "Our findings of targeted vastus lateralis atrophy in severe varus alignment complement previous studies showing muscle-specific changes in OA, suggesting that muscle morphology may be influenced by joint mechanics."   

Limitations (Lines 187-193): The limitations are clearly stated, but a few points could be expanded. For example, you mention that manual tracing during MRI may have introduced variability. Could advanced automated techniques have been considered to reduce this bias? You might also want to acknowledge the potential impact of interobserver variability, despite the use of experienced surgeons.   

Conclusion Lines 195-201: The conclusion is generally well-stated, but the phrase "vastus lateralis atrophy is a contributing factor in varus knee OA" could be revised for clarity. Consider rephrasing to: "Our results suggest that vastus lateralis atrophy may contribute to the severity of varus knee OA." Additionally, the call for longitudinal studies is appropriate, but you might want to specify what such studies should aim to investigate (e.g., the causal relationship between muscle atrophy and knee alignment).   

Line 209: Institutional Review Board Statement: Not applicable Why it's not applicable? according to the study results and methdology it's mandatory to have an IRB approval, please mention the reference number and institution. Also make sure that the article complied with the ethical and procedural requirements of the conduct of sports medicine and exercise science research [Ref]. and refer to this citation (Guelmami N, Ben Ezzeddine L, Hatem G, Trabelsi O, Ben Saad H, Glenn JM, El Omri A, Chalghaf N, Taheri M, Bouassida A, Ben Aissa M. The Ethical Compass: establishing ethical guidelines for research practices in sports medicine and exercise science. International Journal of Sport Studies for Health. 2024 Apr 1;7(2):31-46. https://doi.org/10.5167/uzh-259791)

Comments on the Quality of English Language

Looks fine

Author Response

Comments and Suggestions for Authors

Peer Review Report for "Quadriceps Muscle Assessment in Patients with Advanced Knee Osteoarthritis versus Normal Knee Anatomy: Correlation with Lower Limb Alignment"   

Title: While the title effectively conveys the study's focus, "Quadriceps Muscle Assessment" could be more specific. Consider specifying which components of the quadriceps (e.g., vastus lateralis, vastus intermedius) are being assessed. A more precise title could be, "Assessment of Vastus Lateralis and Intermedius in Advanced Knee Osteoarthritis and Correlation with Lower Limb Alignment."   

è Thank you for your comment. We analyzed not only VL and VI, but also all 4 parts of quadriceps muscle. I understand your comments focused on our results. So we revised our title to "Assessment of Quadriceps muscle in Advanced Knee Osteoarthritis and Correlation with Lower Limb Alignment."   

Abstract: The abstract is generally clear but could benefit from a more succinct presentation of the results and implications. The sentence, "Although it is not clear whether this unbalanced atrophy is a result or a cause of varus knee OA..." could be revised for better clarity. Consider rephrasing to: "The study highlights vastus lateralis atrophy in severe varus alignment, though causality between atrophy and varus knee OA remains uncertain."   

è Thank you for your comment. we revised that sentence according to your comment.

Also add practical implications of this study in the conclusion of the abstract.
è Thank you for your comment. We added practical implications of our results in the conclusion of the abstract. (line 26-27)

Introduction

First paragraph (lines 31-40): The opening sentence, "Total knee arthroplasty is a well-established surgery for patients with advanced knee osteoarthritis (OA)," is somewhat disconnected from the main focus of your study. Consider replacing it with a sentence that directly introduces the quadriceps' role in knee OA, such as: "Quadriceps muscle dysfunction is a common feature in knee osteoarthritis (OA) and is associated with functional decline and disease progression."   
è Thank you for your comment. We revised this sentence according to your comment.

Lines 38-39: You mention that "the roles of the vastus intermedius, medialis, and lateralis in OA remain unclear despite extensive studies." However, later in the introduction, you cite studies that suggest a role for these muscles in OA. This creates a minor contradiction. You might want to clarify that while these muscles have been studied, their specific contributions to OA pathogenesis, particularly in relation to lower limb alignment, remain less established.   
è Thank you for your comment. We revised this sentence according to your comment.

Lines 41-50: The discussion of vastus medialis (VM) atrophy in patellofemoral pain syndrome (PFPS) is informative, but the connection to knee OA needs to be made clearer here. You might want to introduce a linking sentence to bridge the gap between PFPS and knee OA, such as: "While VM atrophy has been associated with patellofemoral pain, its role in knee OA, particularly in conjunction with other quadriceps components, warrants further investigation."  
è Thank you for your comment. We added recommended sentence in the 2nd paragraph of the introduction.

 Lines 46-48: The repetition of the phrase "According to the knee OA study, increased VM size was associated with reduced knee pain and beneficial structural changes at the knee," should be avoided. Consider merging the two instances into one concise sentence: "Increased VM size has been associated with reduced knee pain and structural improvements in knee OA."  
è Thank you for your comment. We deleted the duplicated phrase and revised it according to comment.

Lines 49-50: The sentence, "Conversely, another author suggests that the vastus lateralis (VL) and VM balance might play an important role in OA pathogenesis," is vague. Specify which study you’re referring to and provide a clearer explanation of how this balance could influence OA. A more precise sentence could be: "Pan et al. (2011) suggested that the ratio between vastus lateralis and vastus medialis cross-sectional area may influence the progression of OA."   
è Thank you for your comment. We revised this sentence according to your comment.

Methods   

Line 56: replace "Subjects" by "Participants"   
è Thank you for your comment. We revised this word according to your comment.

Lines 55-65: The inclusion and exclusion criteria are clearly presented. However, the criteria for defining "healthy individuals" (control group) is not entirely clear. Were the control participants free of any knee symptoms, or were they simply classified as KL grade 0 or 1? Clarifying this distinction would strengthen your methodology transparency.  
è Thank you for your comment. As your comment, we added an explanation about healthy control patients.

 Lines 74-82 (Radiographic Evaluations): The description of how participants were divided into alignment groups is well-detailed. However, it would be beneficial to explain why the specific cutoffs of 5° and 10° were chosen for normal, mild, and severe varus. Were these thresholds based on prior literature or clinical guidelines? A brief justification here would improve the rationale.  

è Thank you for your comment. We have referred to Fink B et al.’s reference [11]. Their study divided osteoarthritis group according to 5 degrees. We cited this article.

 Results Table 1 (Demographic Data): The comparison between the OA and control groups is informative, but the significant age difference (approximately 25 years) between the groups is a potential confounder. A possible question for the authors: Why was age not controlled for in the analysis, given its potential influence on muscle atrophy and knee alignment? You may want to mention the need for age-matched controls in future studies to mitigate this issue.   

è Thank you for your comment. we did not perform an age- and sex-matched study. Because we want to compare old-aged OA group and the young healthy group. We added this content in the “limitations”

Lines 121-127 (CSA Ratio): The statement, "no significant difference of the CSA ratio between the knee OA patients and the control group were found," is somewhat surprising given the study's focus. Could the authors provide an explanation for this finding?

è Thank you for your comment. We revised this sentence. “Additionally, we analyzed the ratio of percentages between each component. As a result, no significant difference of the CSA ratio between the knee OA patients and the control group were found (Table 2).”

Specifically, is it possible that the significant reduction in total CSA in the OA group masks potential differences in the relative proportions of the quadriceps components? This discrepancy should be acknowledged and discussed.  
è Thank you for your comment. As we mentioned above, we wanted to compare OA group and young & healthy group. That means the absolute figures may be lower in OA group. Therefore, we thought the relative proportion of the Q muscle would be more meaningful (same as Z-score of BMD)

 Lines 130-135 (Muscle CSA and Alignment): The finding that vastus lateralis CSA was significantly smaller in the severe varus group is well-highlighted. However, the clinical significance of this result is not fully explored. A question for the authors: How might vastus lateralis atrophy contribute to the progression of varus deformity in knee OA? Could strengthening the vastus lateralis potentially mitigate further varus alignment? A brief discussion on the potential therapeutic implications would be valuable.   
è Thank you for your comment. Your question was a key point of our article. We are not sure which came first either, varus deformity or VL atrophy. We guess that VL atrophy is secondary change after varus deformity. Further well-designed study is needed to confirm our hypothesis. We added our opinion in the last paragraph of discussion. Please see 189-192.

Discussion

Lines 143-175 (General Discussion of Results): Your discussion of quadriceps atrophy in knee OA patients is thorough but could benefit from a more focused interpretation of the key findings. For example, the sentence, "Although the ratios of muscle CSA were similar across both groups, OA group participants showed significantly reduced CSA of the total quadriceps," could be more directly tied to the clinical relevance of vastus lateralis atrophy. Consider rephrasing to: "While overall quadriceps CSA was reduced in OA patients, the disproportionate atrophy of the vastus lateralis in the severe varus group suggests a targeted impact of knee alignment on muscle morphology."   
è Thank you for your comment. as your comment, we revised these sentences.

Lines 176-185: The discussion on muscle atrophy and fatty infiltration is relevant but could be better aligned with your own findings. Specifically, you might want to emphasize how your results contribute to the understanding of muscle-specific atrophy in knee OA. For instance, you could add: "Our findings of targeted vastus lateralis atrophy in severe varus alignment complement previous studies showing muscle-specific changes in OA, suggesting that muscle morphology may be influenced by joint mechanics."   
è Thank you for your comment. as your comment, we added these sentences. Please see line 191-195.

Limitations (Lines 187-193): The limitations are clearly stated, but a few points could be expanded. For example, you mention that manual tracing during MRI may have introduced variability. Could advanced automated techniques have been considered to reduce this bias? You might also want to acknowledge the potential impact of interobserver variability, despite the use of experienced surgeons.   
è Thank you for your comment. As your comment, we added this limitation in line 201-202.

Conclusion Lines 195-201: The conclusion is generally well-stated, but the phrase "vastus lateralis atrophy is a contributing factor in varus knee OA" could be revised for clarity. Consider rephrasing to: "Our results suggest that vastus lateralis atrophy may contribute to the severity of varus knee OA." Additionally, the call for longitudinal studies is appropriate, but you might want to specify what such studies should aim to investigate (e.g., the causal relationship between muscle atrophy and knee alignment).   
è Thank you for your comment. As your comment, we modified this conclusion. Please see 211-214.

Line 209: Institutional Review Board Statement: Not applicable Why it's not applicable? according to the study results and methdology it's mandatory to have an IRB approval, please mention the reference number and institution. Also make sure that the article complied with the ethical and procedural requirements of the conduct of sports medicine and exercise science research [Ref]. and refer to this citation (Guelmami N, Ben Ezzeddine L, Hatem G, Trabelsi O, Ben Saad H, Glenn JM, El Omri A, Chalghaf N, Taheri M, Bouassida A, Ben Aissa M. The Ethical Compass: establishing ethical guidelines for research practices in sports medicine and exercise science. International Journal of Sport Studies for Health. 2024 Apr 1;7(2):31-46. https://doi.org/10.5167/uzh-259791)

è Thank you for your comment. We added IRB documents.

Round 2

Reviewer 1 Report

Comments and Suggestions for Authors

Thank you for the revision. All my previous concerns have been addressed.